# The Role of School Social Support and School Social Climate in Dating Violence Victimization Prevention among Adolescents in Europe

**DOI:** 10.3390/ijerph17238935

**Published:** 2020-12-01

**Authors:** Barbara Jankowiak, Sylwia Jaskulska, Belén Sanz-Barbero, Alba Ayala, Jacek Pyżalski, Nicola Bowes, Karen De Claire, Sofia Neves, Joana Topa, Carmen Rodríguez-Blázquez, María Carmen Davó-Blanes, Nicoletta Rosati, María Cinque, Veronica Mocanu, Beatrice Ioan, Iwona Chmura-Rutkowska, Katarzyna Waszyńska, Carmen Vives-Cases

**Affiliations:** 1Faculty of Educational Studies, Adam Mickiewicz University, 61-712 Poznan, Poland; barbara.jankowiak@amu.edu.pl (B.J.); sylwia.jaskulska@amu.edu.pl (S.J.); pyzalski@amu.edu.pl (J.P.); ichmurka@amu.edu.pl (I.C.-R.); katarzyna.waszynska@amu.edu.pl (K.W.); 2National School of Public Health, Carlos III Institute of Health, 28029 Madrid, Spain; 3CIBER of Epidemiology and Public Health (CIBERESP), 28029 Madrid, Spain; carmen.vives@ua.es; 4University Institute on Gender Studies, University Carlos III of Madrid & Research Network on Health Services for Chronic Diseases (REDISSEC), 28903 Madrid, Spain; arwen.alba@gmail.com; 5Department of Applied Psychology, Cardiff Metropolitan University, Cardiff CF52YB, UK; nbowes@cardiffmet.ac.uk (N.B.); kjdeclaire@cardiffmet.ac.uk (K.D.C.); 6Department of Social and Behavioural Sciences, Institute University of Maia, 4475-690 Maia, Portugal; asneves@ismai.pt (S.N.); jtopa@ismai.pt (J.T.); 7CIEG (ISCSP-ULisbon), 1300-663 Lisboa, Portugal; 8National Centre of Epidemiology and CIBERNED, Carlos III Institute of Health, 28029 Madrid, Spain; crodb@isciii.es; 9Department of Community Nursing, Preventive Medicine and Public Health and History of Science, University of Alicante, 03690 Alicante, Spain; mdavo@ua.es; 10Department of Human Sciences, LUMSA University, 00193 Roma, Italy; rosatinicoletta@gmail.com (N.R.); m.cinque@lumsa.it (M.C.); 11Faculty of Medicine, Grigore T. Popa University of Medicine and Pharmacy, 700115 Iasi, Romania; veronica.mocanu@gmail.com (V.M.); ioanbml@yahoo.com (B.I.)

**Keywords:** adolescents, dating violence, school social climate, school social support

## Abstract

The aim of the article is to show the role of school social support and school social climate in dating violence victimization prevention among adolescents in Europe. Study participants were students from secondary schools (age 13–16) in Spain, Italy, Romania, Portugal, Poland and UK. The analysis in this text concern student with dating experience (*n* = 993) (57.2% of girls and 66.5% of boys). School social support was measured by School Social Climate, Factor 1 Scale (CECSCE) and by Student Social Support Scale (CASSS), subscales teachers and classmates. The association between school social support and different types of dating victimization (physical and/or sexual dating violence, control dating violence and fear) was measured by calculating the prevalence ratios and their 95% confidence intervals, estimated by Poisson regression models with robust variance. All the models were adjusted by country and by sociodemographic variables. The results show that the average values of all types of social support are significantly lower in young people who have suffered any type of dating violence or were scared of their partner. The likelihood of suffering physical and/or sexual dating violence decreased when school social support increased [PR (CI 95%): 0.96 (0.92; 0.99)]. In the same way, the likelihood of fear decreased when school social climate increased [PR (CI 95%): 0.98 (0.96; 0.99)].There is an association between school social support and school social climate and experiences of being victim of dating violence among adolescents in Europe. Our results suggest that in the prevention of dating violence building a supportive climate at schools and building/using the support of peers and teachers is important.

## 1. Introduction

Dating violence is defined as commission of violence by one or more members of a couple on another member in the context of dating [1]. According to Centres for Disease Control and Prevention dating violence is defined as physical and/or psychological aggression included in the relationship as well as stalking. It may appear in direct contact or the contact via electronic devices, between current or ex partners [2]. Physical violence includes actions which may lead to injuring the victim (beating, pushing, strangling, shaking, jerking, hitting, using weapon). Emotional violence (frequently referred to as psychological violence) includes the behaviours which cause emotional damage, trauma or fear; these are e.g., threatening with physical violence, controlling the victim’s behaviour, discrediting and intimidating the victim, isolating the victim from family and friends, damaging the victim’s possessions. Sexual dating violence includes getting somebody to have a sexual act against their will (regardless the sexual act takes place) or engaging into sexual activities with a person who is unable to express their consent. Finally, stalking regards to harassment that causes the victim the feeling of fear [3,4].

Dating violence among teenagers is a behaviour that occurs in the context of an intimate relationship between young people [4]. Most studies on aggression in young romantic relationships indicate that both boys and girls may be victims of this kind of violence, although girls are reportedly more likely to experience victimization than boys, also the severity and types of violence can be different due to gender [5,6,7]. In the study conducted by the Centre for Disease Control and Prevention [8], female students reported a significantly higher prevalence physical or sexual dating violence victimization compared to male students (20.9% to 10.4%) [8]. This disparity by sex has also been noted in a representative survey conducted in Canada with in an adolescent population [9]. This study highlighted that psychological violence is the most frequent form of dating violence in youth. Girls were more likely to report all forms of dating violence victimization compared to boys (62.7% vs. 40.5%), with a three-times higher prevalence of sexual dating violence compared to boys (20.2% vs. 5.7%) [9].

In the lights of research on teen dating violence, perpetration and victimization is related to many factors. These is for instance alcohol drinking (persons using various intoxicants including alcohol not only enter violent relationships) [10], but also use alcohol as the way to deal with the fact of being a dating violence victim [11]). Among all these factors, the emotional ones take an important place. For example the research of Foshee et al. [12] proves that some factors like depression, anxiety, anger, marijuana use predicted perpetration by some groups of adolescents (gender, origin and so on) but number of friends using dating violence was a predictor for all groups.

Dating violence affects not only the current functioning of young people, but it has long-term consequences related to their development, affecting how they learn to cope with difficult situations and negatively impacting on their health in a number of different ways [13,14]. Researchers have highlighted the long-term health consequences including: suicidal thoughts, mental illness, psychoactive substance abuse disorders, auto-aggression, depression, eating disorders [4] as well as short-term reduced academic performance and injuries requiring medical care (partner aggression, self-aggression) [15]. Therefore, it is so important for young people to know how to defend themselves against dating violence and, in order to achieve this, use all the resources they possess.

We assume that according to Positive Youth Development model, ecology of development is crucial: family, school and peer environment resources create experiences and bring opportunities to empower development [16]. The people who support the Positive Youth Development model believe that its central place is occupied by immediate environment context. It constitutes an incubator for positive development; here the young may use the resources influencing their development [17]. The factor protecting against violence for instance (including dating violence) is promotion of the access for the teenagers to positive experience, resources, possibilities and development outcomes which are useful both for them and the society. According to this model teenagers are conscious users of their positive development [16].

Our study is based on the assumption that social environment may affect human behaviour and human development, as research indicates that relationships with peers, parents, teachers and school staff may significantly affect the way romantic relationships are shaped [18]. This can explain the relationship between the school climate (which is largely relational [19]), the support of people at school, and dating violence.

Understanding the social support is based on Tardy’s model. What is crucial here is the conceptualization of support through its networks and type. The network regards support sources which are available to people via various relations present in their lives (e.g., support from parents, members of community, friends) [20]. Each type of support reflects a unique set of behaviours and can influence people’s’ attitude and actions in unique ways [21].

The indicators of social support used by us in the research are based on Tardy’s model of social support [22]. Social support at school is defined as emotional, instrumental, informational, and/or appraisal support received from teachers and schoolmates. School climate is a multidimensional construct, referring to the interpersonal relationships, teaching practices, and organizational structures that reflects students, parents, and school personnel’s experiences with the school [23]. Analyzing school reality from the perspective of its climate made capacity for assistance, respect, safety, and comfort as perceived in the school centre [24].

Research results indicate a link between dating violence and various aspects of the school environment [25]. For example, increased school bond, being recognized, and having caring relationships with adults at school are associated with lower level of physical abuse among teenagers [19]. As it is shown in the research, school support is a protective factor for both physical and verbal teen dating violence victimization [26]. School belonging prevents teenagers with adverse childhood experiences from becoming dating violence perpetrators [27]. In adolescence social relations change towards decreasing the meaning of the family and increasing other social groups, especially peer group [28]. Therefore, the relations in school environment may compensate negative experience—also those related to family [29]. Awareness of being a reliable person and an integral part of a community transfers the energy to socially accepted activities. According to the accepted teen development model, positive school relations constitute a factor which protects against violence but experiencing violence, incl. dating violence, may be the reason for withdrawing from relations with people, also schoolmates.

Although the idea that school climate and school support can protect or increase the risk of dating violence may seem simplistic, there may also be more complex relationships between these variables. For example, victims of violence may respond to this negative experience by generalizing social anxiety and because of this, may wrongly assess relationships with other people at school, perceiving greater threats etc. As the results of the research show, experiencing violence weakens ties with school, for example teens who experience online partner violence report lower school bonds [30]. Those who experienced different forms of victimization (e.g., psychological and physical dating abuse) report a lower sense of school belonging [31].

School, or even a classroom as a narrower area, is perceived as an important factor for the socialization of dating violence in adolescents [32]. As for the research, it is related particularly to a relative aspect of school climate and social support. For example, in the research performed by Yang, Wang, Lei, the diagnosed relation between perceived school climate and bullying perpetration was nonsignificant for adolescents with low peers’ support [33]. Research shows that teenagers who experience dating violence seek informal rather than formal help, but if it is formal, the people they turn to are school staff [34]. Supportive atmospheres at school, defined as teacher support perceived by students, is associated with shaping the skill of seeking help in cases of bullying, dating violence and other forms of intimidation. Students who saw their teachers and other school staff as supportive were more likely to favour positive attitudes towards seeking help when they were exposed to intimidation and threats of violence. Getting help from teachers and other school staff when they need it can make them feel positively. In schools with more tangible support, there were fewer differences between girls and boys in attitudes towards seeking help. The findings suggest that school staff efforts to provide a supportive climate are a helpful strategy to prevent bullying and violence [35].

As the research results suggest, both peer [36] and school [37] social contexts are just as important in the prevention of dating violence as the extremely significant factor of family social context is but still there are fewer studies that explore school climate and student development and whilst these are becoming more frequent [38,39] and research on behaviour related to violence and school climate is emerging [19], it lags behind the research on family social context. The impact of different aspects of the school context, individual characteristics, and dating violence are not clear and require further attention and analysis [3]. The aim of the article is input in this knowledge area and checking, in the context of the research based on positive teenager’s development model, the connection between social school support, social school climate and dating violence victimization. We assumed that the development environment rich in support resources and assistance, respect, safety, and comfort, is a factor protecting against dating violence victimization.

This has been one of the project “Lights4Violence” (L4V) assumptions [40], a longitudinal quasi-experimental educational intervention addressed to boys and girls aged between 13 and 17, enrolled in secondary education schools in Alicante (Spain), Rome (Italy), Cardiff (UK), Iasi (Romania), Poznan (Poland) and Matosinhos (Portugal). The project was focused on promoting adolescents’ capabilities to improve their intimate relationships with their peers through different activities that aim to: enable adolescents to acknowledge IPV-related protective factors that are present in themselves, their families, the school and other closed settings, and to know how to properly use them; contribute to education and awareness-raising about the importance of positive interpersonal relationships based on self-esteem and trust; endorse adolescents in challenging sexist and tolerant attitudes towards gender-based violence and dating violence; promote skills to manage problems and conflicts through interpersonal communication, mediation and negotiation among youth, and empower young people to claim their rights and those of their peers to be held in esteem and to protect themselves from at-risk or abusive relationships [41].

Lights4Violence project gave us the opportunity to identify how socioeconomic characteristic, personal experiences, resources, both personal and environmental, and competencies are associated with experiences of dating violence. This study explores the relationship between school climate, school social support and exposure to dating violence (physical, sexual and emotional). In this article we concentrate only on school context but still being aware that positive teen development model assumes empowering development in wide ecological context.

## 2. Materials and Methods

### 2.1. Design

The study has a cross-sectional design. The data was gathered in an adolescent at the baseline stage of their engagement in the “Lights4Violence” activities, before their participation in the workshops and film classes described above. The data was collected using an online questionnaire distributed at schools in participating countries. The respondents participated during two technology classrooms, preferably before and after the morning break, with a maximum duration of an hour and 15 min. The on-line questionnaire included demographic variables, socioeconomic variables, and experience of dating, school social support and other scales defined by the project ‘Lights4Violence’.

The data was gathered in 12 schools between October 2018 and February 2019. The selection of schools was carried out by contacting different secondary education Centres from the city with non-random inclusion of institutions. The program content was presented, and the opportunity to participate was offered to the school headmasters. Participation was offered to all the students of the classes selected. The percentage of participation was 98.78 percent.

All the information gathered by the Project partners and beneficiaries was confidential. The participation of the target groups was voluntary and required the permission of the ethical committee of each university and a signed informed consent document from the school, headmasters, parents and students.

The Lights4Violence Project protocol was approved by the ethical committee of the University of Alicante, Instituto Universitário da Maia/Maiêutica Cooperativa de Ensino Superior CRL. Maia, Universitatea de Medicina si Farmacie Grigore T. Popa and Adam Mickiewicz University. Waivers were obtained from the Libera Universita Maria SS. Assunta of Rome and the Cardiff Metropolitan University. These ethics approvals/waivers covered the individual schools where the intervention was performed. It was also registered in ClinicalTrials.gov by the coordinator (Clinicaltrials.gov: NCT03411564. Unique Protocol ID: 776905. Date registered: 18 January 2018).

### 2.2. Participants

We recruited 1555 participants between 13 and 16 from secondary schools in Alicante, Spain (130 girls and 125 boys), Rome, Italy (206 girls and 79 boys), Iasi, Romania (214 girls and 129 boys), Matosinhos, Portugal (125 girls and 134 boys), Poznań, Poland (135 girls and 55 boys) and Cardiff, UK (112 girls and 92 boys). The analyses in this text concern students with dating experience (*n* = 993) (57.2% of girls and 66.5% of boys). A statistical power analysis was performed for sample size estimation, based on data from previously published random-effects meta-analysis of 23 studies about school-based interventions aimed to prevent violence and negative attitudes in teen dating relationships [42].

### 2.3. Measures

#### 2.3.1. Predicted Variables

In this study, the predicted variable was the exposure to dating violence and this was measured across three aspects; physical, sexual and control dating violence. We also analysed the feeling of “fear” as a proxy of violence.

Those who had been in a dating relationship were asked:

Physical dating violence: “Has anyone that you have ever been on a date with physically hurt you in any way? (For example, slapped you, kicked you, pushed, grabbed, or shoved you)”;

Sexual dating violence: “Has the person that you have been on a date with ever attempted to force or force you to take part in any form of sexual activity when you did not want it?”;

Control dating violence: “Has the person that you have been on a date with ever tried to control your daily activities, for example, who you could talk with, where you could go, how to dress, check your mobile phone etc.?”.

Fear: “Has the person that you have been on a date with ever threatened you or made you feel so in any way?”

#### 2.3.2. Predictor Variables

In this study, we considered two predictor variables collected by the Student Social Support Scale (CASSS), subscales teachers and classmates and by the School Social Climate, Factor 1 (CECSCE).

##### Student Social Support Scale (CASSS)

The Student Social Support Scale subscales teachers and classmates, assesses the student’s perceived emotional, appraisal, informational, and instrumental social support from two areas: teachers and classmates. It includes 12 items each subscale with 6 Likert-type response categories that range from never to always. A higher score indicates greater social support. Students rate each behaviour on two dimensions: availability (6-point rating scale) and importance (3-point rating scale) [22]. For this study, we only analyzed the results of the frequency subscale because the trend of both dimensions related to dependent variables and co-variables was very similar. In our study, CASSS showed satisfactory internal consistency (Cronbach’s alpha) ranged from 0.94 (Poland) to 0.97 (United Kingdom and Portugal). CASSS items examples are “My teacher… helps me solve problems by giving me information… explains things that I don’t understand”, “My classmates…, nicely tell me when I make mistakes… give me good advice… spend time doing things with me”.

##### School Social Climate Questionnaire (CECSCE)

The CECSCE is a questionnaire that assesses school social climate. It is made up of items from The California School Climate and Safety Survey. The CECSCE displays a stable factorial structure in two social climate factors: 1) relative to the school and 2) relative to the teaching staff. In this project we will use factor 1 only. The 8 items that saturate the first factor are indicative of the capacity for assistance, respect, safety and comfort, as perceived in the school centre. Items are rated on a 5-point Likert-type scale, from strongly agree to strongly disagree [43]. Cronbach’s alpha of CESCE ranged from 0.73 (Italy) to 0.85 (Portugal and United Kingdom) in the present study. CECSCE items examples ae: “At this school… the students are really motivated to learn… students of all races and ethnic groups are respected.

#### 2.3.3. Covariates

Adjustment demographic and socioeconomic variables collected including: sex (male; female; other sex −1.2% of the “other sex” category was considered as lost/missing value-), age (continues variable), father employment status (paid work/other). As in some countries it was common for mothers to work exclusively as housewives, we include the father’s employment status as an indicator of the household’s socioeconomic status. All the models were adjusted by country (Poland, Portugal, Spain, Italy, Romania and United Kingdom.

#### 2.3.4. Statistical Analyses

A descriptive analysis of the sample was carried out for each of the variables included in the study. We described the mean and the standard distribution of our independent quantitative variables and percentage distribution in qualitative variables. We describe the prevalence of different types of dating violence (physical, sexual and control) and fear, in the whole sample and stratified by sex. Differences between girls and boys prevalence were estimated by Chi–square test. The mean differences on social support (CASSS subscales and CECSCE) by different types of dating violence and fear were estimating using the Student’s *t*-test and analysis of variance.

The association of the exposure to physical and/or sexual dating violence, control dating violence and fear was measured by calculating the prevalence ratios (PR) and their 95% confidence intervals (CI), estimated by Poisson regression models with robust variance. In order to avoid biases due to erroneous classification, the regression analysis considered women exposed to IPV in comparison to women who had never suffered any type of violence. All the models were adjusted by sociodemografic variables previously described and by country.

## 3. Results

### 3.1. Descriptive Analysis of the Sample

Table 1 shows descriptive analysis of the sample. In the research sample, 56.5% are girls and 43.5% boys. In most of the cases the father of the respondents has paid work (89.4%). 10.6% have no paid job (homemaker, unemployed, pensioner, student). Mean respondent’s age is 14.3 years (SD = 1.50). Mean for school climate is 27.5 (min. 8, max. 40, SD = 6.04) for teacher support 49.8 (min. 12, max. 72, SD = 12.95), classmates support 48.5 (min. 12, max. 72, SD = 12.80).

### 3.2. Prevalence of Dating Violence

The obtained data show that the 20.5% of girls and 18.7% of boys that were or have ever dating, reported that they had suffered dating violence. The most common type of violence experienced by young people is emotional violence consisting in controlling the daily activities of a partner (23.1%). The 25.1% of girls and 19.6% of boys experience this form of violence. The sense of threat caused by the partner’s concerns 7.9% students. Regard to physical violence in relationships, it affects 8.8% young people, sexual violence 10.9% of girls and 6.1% of boys. There are statistically significant differences between boys and girls in being a victim of sexual (*p* = 0.009) and emotional abuse in terms of partner’s control of daily activities (*p* = 0.043). In both cases, girls are more likely to be victims of violence than boys (Table 2.).

### 3.3. School Social Support and School Social Climate

Table 3 shows the differences in means in different types of school social support according to the exposure of dating violence. The results obtained show that the average values of all types of social support are significantly lower in young people who have suffered any type of dating violence or were scared of their partner.

Table 4 shows the robust Poisson regression models. These models identify the association between the different school social support analysed (CECSCE, CASSS teacher, CASSS classmates) and the different types of dating violence/fear. Once we adjusted by sociodemographic variables and country, the likelihood of suffering physical and/or sexual dating violence decreased when CECSCE increased [PR (CI 95%): 0.96 (0.92; 0.99)]. In the same way, the likelihood of fear decreased when CASSS classmates increased [PR (CI 95%): 0.98 (0.96; 0.99)].

## 4. Discussion

The school’s climate has been proven to be a really important consideration when exploring physical, emotional and sexual violence within teenage dating relationships. Our study demonstrated relations of some dimensions of school climate and experience of dating violence victimization. The dependency may spread in various directions. Positive school experience may for instance protect against entering relations with persons who deviate from positive experience in relations whilst negative experience of after-school life areas make a student notice the potential of school environment or isolate from it. School environment may also give the resources protecting against entering relations which threaten with violence. Good school climate can be seen as a generalized sense of security at school. This is an important consideration for schools, improving the climate in a school may be an important means of supporting children against entering into relationships where their physical and sexual well-being are threatened.

Our results are consistent with previous studies. A high level of identification and bond with the school, as well as the perception of one’s own agency and security in school relationships were significantly associated with lower chances of adolescent relationship abuse [19].

The results of previous studies also show that a good school climate protects students from premature sexual activity, reduces the risk of unwanted pregnancy and sexually transmitted diseases [42]. A safe climate at school is probably conducive to conversations on any topic, also related to sex education in a broad sense, including the topic of sexual abuse. According to the results of previous studies on school climate, it can also be said that it generally counteracts aggression at school [44], including sexual and physical aggression in intimate relationships, which are often perpetrated by peers attending the same school and, largely, perpetrated on school premises.

Building a safe climate at home and at school is a protective factor [45], which is also confirmed in relation to the school by our analyses.

Research results indicate that the importance of support from teachers and classmates in experiencing dating violence should be considered because those who have not experienced physical, sexual or emotional abuse perceive higher support at school than those who have experienced violence or this violence in particular or have other negative experience in after-school life and are not able to notice and make the advantage of school resources. According to the regression analysis, the support of colleagues has particular importance for fear, as a proxy of violence.

Relationships with classmates at school protect in the non-physical but more discrete sphere, the emotional one. One of the more common conclusions from previous studies, including those dealing with women experiencing violence in intimate relationships, is that social support prevents or alleviates the psychological effects of being a victim of violence—for example, greater social support reduces depression and anxiety of victims [46]. Conclusions from previous studies show also that social support serves as a mediator of the relationship between dating violence victimization and psychological well-being [47]. Peer support can help to protect, above all, from emotional abuse by mobilizing one’s own psychological resources.

In the light of research results, social factors are a potential protection for abused women against the development of anxiety and depression, but they prove to be less effective in protecting them from becoming a victim during their lifetime [48]. Social support, even if it does not protect against entering a violent intimate relation, helps to cope with its psychological consequences [49]. This explanation is supported by our results, supportive relationships with peers are protective against fearing and a sense of danger in the relationship.

In the lights of our research-like the previous ones-it is necessary to create not only preventive programs that raise awareness of teen dating violence at schools [50], but also the programs concentrating on forming school climate and school social support. It also becomes crucial to strongly indicate that these resources may only be used in dating violence prophylaxis [51].

In interpreting our results, it is necessary to consider some limitations. The sample size does not allow us to generalize the study results to the population of each country because the sample size was small. The sample size was calculated to have enough statistical power to analyse the results as a whole. It should also be mentioned that some information related to sociodemographic characteristics were lost, because the adolescents declined to provide that information. The information lost was related to parents’ education level, because they did not know this information about them, so a high percentage of the answers were marked as “don’t know”. The perception of having been exposed to dating violence could be different depending on the cultural context of the students. To address this, our models have been adjusted by country, but there may be residual confusion. In this case, in the countries that identify least intimate partner violence, the effect of school social support on dating violence could be underestimated.

## 5. Conclusions

There is an association between school social support and school social climate and experiences of dating violence among adolescents in Europe. Despite of the limitations, we can say that school climate and the support of peers are factors that protect against dating violence. Regarding the above, the programs against dating violence victimization, which are aimed at schools, should provide not only the knowledge on this violence and social skills but also the work in the area of school climate and relations which may become a source of support.

## Figures and Tables

**Table 1 ijerph-17-08935-t001:** Descriptive analysis of the sample (*n* = 993).

	***N***	**%**
Sex		
Girls	551	56.5
Boys	424	43.5
Age		
≤13 years	310	31.22
14–15 years	468	47.13
>15 years	215	21.65
Country		
Italy	202	20.34
Poland	108	10.88
Portugal	179	18.03
Romania	190	19.13
Spain	184	18.53
United Kingdom	130	13.09
Father’s employ		
No paid work (homemaker, unemployed, pensioner, student)	98	10.6
Paid work/freelance	826	89.4
	Mean	SD
Age	14.3	1.5
CECSCE	27.5	6.04
CASSS teacher	49.8	12.95
CASSS classmates	48.5	12.8

**Table 2 ijerph-17-08935-t002:** Prevalence of dating violence victimization.

Dating Violence	Total		Girls		Boys		Effect Size (Cramer’s V)	*p*-Value *
	*n* = 993		*n* = 551		*n* = 424			
	*N*	%	*N*	%	*N*	%		
Physical	87	8.8	51	9.3	31	7.3	0.116	0.278
Sexual	92	9.3	60	10.9	26	6.1	0.271	0.009
Control	229	23.1	138	25.1	83	19.6	0.134	0.043
Fear	78	7.9	45	8.2	23	5.4	0.189	0.096

* Differences by sex chi-square.

**Table 3 ijerph-17-08935-t003:** Means of social school support and school climate by dating violence victimization.

Dating Violence	CECSCE	CASSS Teacher	CASSS Classmates
	Mean (SD)	Mean (SD)	Mean (SD)
Physical			
Yes	25.0 (6.6)	44.8 (13.7)	45.5 (12.9)
No	27.7 (5.9)	50.3 (12.8)	48.8 (12.8)
*p*-value *	<0.001	<0.001	0.020
*ES*	0.43	0.41	0.26
Sexual			
Yes	25.0 (6.2)	44.3 (13.0)	43.4 (14.1)
No	27.7 (6.0)	50.4 (12.8)	49.0 (12.6)
*p*-value *	<0.001	<0.001	<0.001
*ES*	0.44	0.47	0.42
Control			
Yes	26.6 (6.1)	46.1 (12.9)	45.6 (12.2)
No	27.8 (6.0)	51.0 (12.8)	49.4 (12.8)
*p*-value *	0.01	<0.001	<0.001
*ES*	0.20	0.38	0.30
Fear			
Yes	24.7 (6.3)	43.5 (14.0)	40.8 (13.6)
No	27.7 (6.0)	50.4 (12.7)	49.2 (12.5)
*p*-value *	<0.001	<0.001	<0.001
*ES*	0.49	0.52	0.64

SD: Standard deviation; ES: Effect size (Cohen’s d); * difference Student *t*-test.

**Table 4 ijerph-17-08935-t004:** Poisson regression with robust variance. Association between school social support and the different types of dating violence/fear.

	Crude				Adjusted *			
	PR	CI 95%		*p*-Value	PR	CI 95%		*p*-Value
Physical and/or sexual								
CECSCE	0.94	0.92	0.96	<0.001	0.96	0.92	0.99	0.008
CASSS teacher	0.97	0.96	0.98	<0.001	0.99	0.97	1.00	0.132
CASSS classmates	0.97	0.96	0.99	<0.001	0.99	0.98	1.00	0.158
Control								
CECSCE	0.97	0.96	0.99	0.002	1.00	0.97	1.02	0.724
CASSS teacher	0.98	0.97	0.99	<0.001	0.99	0.98	1.00	0.173
CASSS classmates	0.98	0.97	0.99	<0.001	1.00	0.99	1.01	0.449
Fear								
CECSCE	0.93	0.91	0.96	<0.001	0.96	0.92	1.02	0.165
CASSS teacher	0.96	0.95	0.98	<0.001	1.00	0.97	1.02	0.740
CASSS classmates	0.95	0.94	0.97	<0.001	0.98	0.96	0.99	0.034

* Adjusted by age, sex, country, and socioeconomic variable; PR: prevalence ratio; CI: confidence interval at 95% level.

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
