# Peer review of "The Role of School Social Support and School Social Climate in Dating Violence Victimization Prevention among Adolescents in Europe"

_ijerph, 2020, doi:10.3390/ijerph17238935_

Round 1

Reviewer 1 Report

My concerns with this paper relate primarily to the logic and quality of its argument and the conclusions reached. Other issues that need to be addressed but could be done so relatively easily, include:

  • A lack of definitional clarity regarding key terms or concepts - such as 'social support', 'school social support' and 'school climate' - is also an issue that could be addressed.
  • The authors make frequent references to 'CECSCE' and 'CASS' but there is no indication of what these mean or stand for. 

Overall, the authors have a tendency to overstate in a way that either/both simplifies complex ideas and issues and/or isn't warranted based on the works cited.

  • For example, page 2, lines 55-58:  One reference is provided in relation to the statement "Most studies on aggression in young romantic relationships indicate that both boys and girls ...[etc]". Further, the source cited relates to a specific context, age group/cohort, country, etc. so is not an appropriate reference (on its own) to substantiate the authors' claim.
  • Similarly, the authors' discussion, in general, lacks depth and precision. For example, much of their argument seems to be based on the idea that "school belonging prevents teenagers with adverse childhood experiences
    from becoming dating violence perpetrators" (page 2, lines 80-81). There is little-no explanation of this claim (i.e. how so? why?) nor do the authors explore this or distinguish between cause, correlation and effect. eg perhaps belonging is not causative but rather teenagers who perpetrate dating violence are less likely to experience school belonging .. (etc). A similar observation could be made regarding statements such as "Students who saw their teachers and other school staff as supportive were more likely to favor positive attitudes towards seeking help ..." (p. 2, lines 94-96). For other examples, see page 7 (lines 238-239 and 258-260).  
  • The reference to "skills inherent to active and participative citizenship which leads them to assume the respect for democratic and human rights" (p. 3, lines 101/102) seems out of place in this paragraph and doesn't make sense in this context. 
  • Results section, p. 5 - some explanation is necessary in relation to the inclusion of employment status data for fathers only (but not mothers).  
  • Greater precision is needed in wording and written expression - for example, sentences like "The opportunity to protect oneself against abuse and sexual violence is ..." (p. 7, lines 249/50) are clumsy and their meaning is unclear. Relatedly the paragraph at top of page 8 is confusing  due to poor expression and/or word choice.
  • The limitations (page 8, lines 278 - 280) could do with further thought. Eg the generalisability of the study is likely to relate to its cultural context, not sample size.

Finally, in the conclusion (page 8), the final sentence - "The anti-violence atmosphere of school prevents the experience of dating violence by providing a sense of security, flow of knowledge and support" is not supported by the results and discussion sections as they appear in this paper. If indeed this is a reasonable conclusion to reach (based on the data itself), the authors will need to strengthen their account of this - and of their interpretation and analysis - in order to convince the reader.

I hope this feedback is useful to the authors in refining their paper for publication.

Reviewer 2 Report

The present manuscript analyses the links between some school variables and dating violence victimization in adolescents. Teen dating violence is a serious health problem, and it is an important topic of research. Nevertheless, this manuscript presents serious theoretical and methodological deficiencies.

Abstract

The abstract should be structured in parts, but these parts should not be numbered. In addition, it is necessary to include a short introduction (at least one sentence) about the topic of study and its current relevance. Acronyms should not be included in the abstract. The authors should better clarify the main results of the study and the main conclusions. In general, the abstract is confusing. For example, regarding the sample, authors write (lines 36-37): “The analysis in this text concerns student with dating experience (n=993) (57.2% of girls and 66.5% of boys)”. Although reader, perhaps, may suppose that these data is referred to the percentage of boys and girls who had a partner, compared to the initial sample, this information here is confusing. The information usually indicated in the abstract is only how many boys (%) and girls (%) were in the final sample (being the total percentage 100). Abstract should be reviewed to avoid confusion for readers.

Introduction

Authors should expand the review of previous studies on teen dating violence. The review is scarce. Moreover, it is necessary to include a specific definition of teen dating violence, explain main characteristics (for example, main differences from dating violence in adult partners) and describe more the different types (physical, psychological, sexual, relational, cyber-abuse). Authors may review, for example, previous studies of Wolfe, Baker, Foshee, Fernández-Fuertes, Leen or Shorey, about teen dating violence. Results of previous studies on the prevalence of different types of teen dating violence needs to be included.

This study is focused on dating violence victimization, and it is not analyzed dating violence perpetration. However, this is not clear in the title of the manuscript. Title should include “dating violence victimization”. More studies on dating violence victimization should be included in Introduction.

It is necessary to provide a clearer definition of school social climate. The authors should extend the review of previous studies on the school social climate, and its relationships with dating violence. There are numerous studies on the influence of peers on dating violence, and previous studies linking school climate and peer violence, and teen dating violence. For example:

Beckmann, L., Bergamnn, M.C., Krieg, Y., & Kliem, S. (2019). Associations Between Classroom Normative Climate and the Perpetration of Teen Dating Violence Among Secondary School Students. Journal of Interpersonal Violence, 1-31. https://doi.org/10.1177/0886260519888207

Yang, J., Wang, X. & Lei, L. (2020). Perceived school climate and adolescents’ bullying perpetration: A moderated mediation model of moral disengagement and peers’ defending. Children and Youth Services Review, 109, 104716

In Introduction section, there is not a theoretical framework from which to propose the objectives of the study, justify hypotheses, and analyze the results. For example, theories on adolescent development or ecological model might be suitable.

Lines 108-120: The objective of this study is not related to the Lights4Violence project (for example, to analyze its effectiveness). Therefore, the description of this project is not relevant in the Introduction. By contrast, authors should focus more on the variables they are going to analyze, and better justify their objective and their hypotheses.

The objectives must be more specific. It is also necessary to provide justified hypotheses based on previous studies.

Methods

Line 126: “Data was collected using an on-line questionnaire”. It is necessary to provide more information on the procedure followed to obtain the data: how were the adolescents informed about the study and its objectives? how was obtained the parental consent?, Were the teachers present at the time of filling in the data?

Participants.

The characteristics of the final sample should be described in more detail: percentage of boys and girls, percentage of students from each country, frequency, and percentage by age, and by school grade. Include more information on the characteristics of schools: size of the schools, rural-urban context, ...

Measures.

It is more suitable to indicate "predictor variables" rather than "independent variables” since these variables are not manipulated by the researcher. The same in relation to “dependent variables”.

The measurement of the predicted variables is weak, since only one item is used for each variable. It would be convenient to use several items on a scale with previously validated psychometric properties.

In general, regarding all the instruments used, it is necessary to provide more information on their psychometric properties: reliability, factor analysis ...

Results

Line 204: “Most of the fathers of the respondents”: fathers or parents? it is also included mothers or only information about fathers?

Table 2 and Table 3: Information on effect size needs to be included.

Lines 226-230: More information on the type of regression analysis performed should be included, justifying the type of analysis used. Provide a more detailed explanation of the results obtained and the predictive value of each of the predictor variables (for example, beta value).

Explain why it has been considered jointly physical and/or sexual in regression analysis.

Discussion

In lines 237-239: “Our study demonstrated that participants who rated school climate more positively were less likely to experience physical and sexual violence within dating relationships”. However, this conclusion cannot be made with the data obtained in this study. Since it is not a longitudinal study, caution should be exercised about the results. Furthermore, only some variables have shown a predictive capacity for some types of dating violence. This conclusion is too broad and not sufficiently supported by the results obtained.

Moreover, it should also be considered that other variables could explain this relationship, without this implying a causality of the perceived support and the perceived climate on dating violence. For example, adolescents with greater difficulties in social and school adaptation (less social skills, lower self-esteem, more social anxiety) may have more difficulties both in relationships with their classmates and teachers and in relationships with their partners.

Although the school climate is a relevant variable to explain violence in adolescents, as many previous studies have shown, the results of the present study do not allow us to indicate causality regarding the relationship between the analyzed variables.

Reviewer 3 Report

The paper explored an important topic and may contribute to the body of knowledge. Please find below comments for consideration

Page 1. Line 33-34: What do CECSCE and CASS mean? Not provide acronym in the abstract.

Introduction: Provide a clear definition of school social support and factors related to dating violence.

Materials and methods:

  • Provide more information about questionnaire used.
  • Provide information about inclusion and exclusion criteria.

Line 33: Do not use acronyms in the abstract.

Line 53-54: Provide a reference at the end of the sentence.

Line 55-58: Provide information about percentages of boys and girls who are victims of dating violence and what is the most common form of dating violence between young people.

Line 66: Why are peer and school social contexts important in the prevention of dating violence?

Line 73-75: How do peers, parents,teachers and school staff influence romantic relationship?

Line 81: Define school support.

Line 118: Provide more information about lights4violence project.

Line 141-148: Provide information about inclusion and exclusion criteria.

Line 204: Fathers o parents?

Line 236-242: Provide references for statements.

Line 257-261: Provide references for statements.

Line 262-266: It’s a sentence too long.

Line 278: Another limitation could be the exposure of teenager to violence in the family or single parents.

Line 290: Not provide references in the conclusion.

Round 2

Reviewer 2 Report

All the questions required have been adequately answered, and the revised version of this manuscript has improved greatly. In my opinion, the revised version of the manuscript can be published with only some minor changes:

It is necessary to review all the manuscript to eliminate the letters A, B, C ... at the end of some sentences. It is not clear what these letters mean? Are they references? If they are references, it would be necessary to indicate the correct reference. They should be reviewed and removed, for example, on the following lines:

Line 54: “…in the context of dating [A].”

Line 57: “…current or ex partners [B].”

Line 65: “… victim the feeling of fear. [C] [1].”

Line 70: “…can be different due to gender [2, J1, K2, L]”

…lines 96, 98, 102, 110, 111…

Line 147: “… of dating violence in adolescents [L9].”

Line 147: “… of dating violence in adolescents [L9]”.

Line 59: The authors write: “Emotional violence (frequently referred to as mental violence) includes …”. Nevertheless, this type of dating violence (emotional violence) is more frequently referred to as “psychological violence” and not “mental violence”. In fact, the authors use “psychological violence” in line 74 to refer to this type of violence. It is advisable to change in line 59 “mental violence” by “psychological violence”.

Line 82: It is necessary to change “the research of Foshee, V. A., Reyes, H. L., Ennett, S. T. proves…” to “the research of Foshee et al. proves…”

The manuscript must be conformed to the style guidelines of this journal. Thus, footnotes should be removed from the entire manuscript.

In addition, it is necessary to review the number of all references throughout the manuscript, and include and renumber all references at the end of the manuscript (references section).

Line 94: “…according to positive teen development model”. This theoretical model is more often cited as “Positive Youth Development model” (PYD model). It is advised to refer to this model as "Positive Youth Development model". Also, review the translation in line 96: “empower development [D]. The followers of positive teen development model”: ¿followers?

Line 132: Review “[przypis z rozwojówki].” What that means?

Line 156: Review the translation in “And as feedback”

Line 195: Review the sentences “The study hasa cross-sectional design. The data was gathered inan adolescents at the baseline” (there are some typos).

Lines 230 and 231: Change “outcomes” by “Predicted variables”

Lines 248-264: Instruments: It is advisable to include some examples of items

Line 311: There is no title in Table 3

Author Response

Dear Reviewer,

thank you for your comments and suggestions. We accepted all of them and put it in the text. Our answers below.

All the questions required have been adequately answered, and the revised version of this manuscript has improved greatly. In my opinion, the revised version of the manuscript can be published with only some minor changes:

It is necessary to review all the manuscript to eliminate the letters A, B, C ... at the end of some sentences. It is not clear what these letters mean? Are they references? If they are references, it would be necessary to indicate the correct reference. They should be reviewed and removed, for example, on the following lines:

Line 54: “…in the context of dating [A].”

Line 57: “…current or ex partners [B].”

Line 65: “… victim the feeling of fear. [C] [1].”

Line 70: “…can be different due to gender [2, J1, K2, L]”

…lines 96, 98, 102, 110, 111…

Line 147: “… of dating violence in adolescents [L9].”

Line 147: “… of dating violence in adolescents [L9]”.

Thank you for your comment.  In this version we have changed the letters by the references.

Line 59: The authors write: “Emotional violence (frequently referred to as mental violence) includes …”. Nevertheless, this type of dating violence (emotional violence) is more frequently referred to as “psychological violence” and not “mental violence”. In fact, the authors use “psychological violence” in line 74 to refer to this type of violence. It is advisable to change in line 59 “mental violence” by “psychological violence”.

 Thank you for your comment. We are agreed with you. We have changed “mental violence” to “psychological violence”

Line 82: It is necessary to change “the research of Foshee, V. A., Reyes, H. L., Ennett, S. T. proves…” to “the research of Foshee et al. proves…”

Thank you. We did the change.

The manuscript must be conformed to the style guidelines of this journal. Thus, footnotes should be removed from the entire manuscript.

Thank you. In this version there aren´t footnotes.

In addition, it is necessary to review the number of all references throughout the manuscript, and include and renumber all references at the end of the manuscript (references section).

Thank you. In this version all the references are correctly renumbered.

Line 94: “…according to positive teen development model”. This theoretical model is more often cited as “Positive Youth Development model” (PYD model). It is advised to refer to this model as "Positive Youth Development model". Also, review the translation in line 96: “empower development [D]. The followers of positive teen development model”: ¿followers?

Thank you for your comment. Now you can read: The people who support the Positive Youth Development model….”

Line 132: Review “[przypis z rozwojówki].” What that means?

Thank you. We have delated it.

Line 156: Review the translation in “And as feedback”

Thank you for your comment. Now you can read:

“Getting help from teachers and other school staff when they need it can make them feel positively.”

Line 195: Review the sentences “The study hasa cross-sectional design. The data was gathered inan adolescents at the baseline” (there are some typos).

Thank you for your comment. Now you can read:

“The data was gathered in an adolescent at the baseline”

Lines 230 and 231: Change “outcomes” by “Predicted variables”

Thank you for your comment. We did the change.

Lines 248-264: Instruments: It is advisable to include some examples of items.

Thank you for your comment. Now you can read:

“CASSS items examples are “My teacher.., helps me solve problems by giving me information… explains things that I don’t understand”, “My classmates…, nicely tell me when I make mistakes….. give me good advice….spend time doing things with me”.”

“CECSCE items examples ae: “At this school… the students are really motivated to learn… students of all races and ethnic groups are respected.”

Line 311: There is no title in Table 3

Thank you for your comment. Now you can read

Table 3. Means of social school support and school climate by dating violence
